# Uncovering Dryland Woody Dynamics Using Optical, Microwave, and Field Data—Prolonged Above-Average Rainfall Paradoxically Contributes to Woody Plant Die-Off in the Western Sahel

**Paulo N. Bernardino** [1,2,*] , **Martin Brandt** [3] , **Wanda De Keersmaecker** [2], **Stéphanie Horion** [3], **Rasmus Fensholt** [3], **Ilié Storms** [1], **Jean-Pierre Wigneron** [4], **Jan Verbesselt** [2,†] and **Ben Somers** [1,†]

1  Division Forest, Nature and Landscape, KU Leuven, Celestijnenlaan 200E, 3001 Heverlee, Belgium; ilie.storms@kuleuven.be (I.S.); ben.somers@kuleuven.be (B.S.)
2  Laboratory of Geo-Information Science and Remote Sensing, Wageningen University, Droevendaalsesteeg 3, 6708 PB Wageningen, The Netherlands; wanda.dekeersmaecker@wur.nl (W.D.K.); jan.verbesselt@wur.nl (J.V.)
3  Department of Geosciences and Natural Resource Management (IGN), University of Copenhagen, Øster Voldgade 10, K 1350 Copenhagen, Denmark; mabr@ign.ku.dk (M.B.); stephanie.horion@ign.ku.dk (S.H.); rf@ign.ku.dk (R.F.)
4  Interactions Sol Plante Atmosphére (ISPA), Unité Mixte de Recherche 1391, Institut National de la Recherche Agronomique (INRA), CS 20032, CEDEX, 33882 Villenave d'Ornon, France; jean-pierre.wigneron@inra.fr
*  Correspondence: paulo.nbernardino@gmail.com; Tel.: +32-496-895300
†  Joint last authors.

**Abstract:** Dryland ecosystems are frequently struck by droughts. Yet, woody vegetation is often able to recover from mortality events once precipitation returns to pre-drought conditions. Climate change, however, may impact woody vegetation resilience due to more extreme and frequent droughts. Thus, better understanding how woody vegetation responds to drought events is essential. We used a phenology-based remote sensing approach coupled with field data to estimate the severity and recovery rates of a large scale die-off event that occurred in 2014–2015 in Senegal. Novel low (L-band) and high-frequency (Ku-band) passive microwave vegetation optical depth (VOD), and optical MODIS data, were used to estimate woody vegetation dynamics. The relative importance of soil, human-pressure, and before-drought vegetation dynamics influencing the woody vegetation response to the drought were assessed. The die-off in 2014–2015 represented the highest dry season VOD drop for the studied period (1989–2017), even though the 2014 drought was not as severe as the droughts in the 1980s and 1990s. The spatially explicit Die-off Severity Index derived in this study, at 500 m resolution, highlights woody plants mortality in the study area. Soil physical characteristics highly affected die-off severity and post-disturbance recovery, but pre-drought biomass accumulation (i.e., in areas that benefited from above-normal rainfall conditions before the 2014 drought) was the most important variable in explaining die-off severity. This study provides new evidence supporting a better understanding of the "greening Sahel", suggesting that a sudden increase in woody vegetation biomass does not necessarily imply a stable ecosystem recovery from the droughts in the 1980s. Instead, prolonged above-normal rainfall conditions prior to a drought may result in the accumulation of woody biomass, creating the basis for potentially large-scale woody vegetation die-off events due to even moderate dry spells.

**Keywords:** drylands; drought; NDVI; passive microwave; time series; vegetation optical depth; woody vegetation dynamics

## 1. Introduction

Dryland ecosystems support the livelihood of almost one-third of the human population and dominate inter-annual dynamics in the global land carbon sink [1–3]. Being one of the largest dryland areas globally, the Sahel extends over around three million km$^2$ on the African continent with a human population of more than 100 million people [4,5]. The water dependency of the Sahelian vegetation [6,7] makes it susceptible to changes in rainfall patterns, with a direct impact on the livelihood of the local population [3]. This does not only apply for the herbaceous vegetation, which serves as fodder for livestock, but also for the woody vegetation which serves, for example, as a source of fuelwood, construction material, fruits, shelter, and livestock fodder [3]. Drought-induced tree and shrub mortality can thus have a considerable impact on livelihoods, as it was observed after the Sahel drought in the 1980s [8].

After two extremely dry periods in the Sahel in the 1970s and 1980s, herbaceous and woody vegetation showed strong signs of recovery [9–12], also known as the "greening Sahel". However, focusing on the overall greening trend over the whole Sahel may undervalue degradation processes happening at smaller spatial scales and the role of short-term vegetation dynamics on the greening trend [11,13–15]. Although negative rainfall anomalies are recurrent in the Sahel and vegetation is able to withstand/recover from sporadic droughts [10,16,17], more extreme and frequent droughts are expected in the coming decades, increasing the risk of tree mortality and of a potential reversal in the "greening Sahel" scenario [3,18,19]. Moreover, besides drought duration and intensity, the ability of ecosystems to cope with drought depends on several other factors [20–22], and soil properties have shown to be important determinants of woody vegetation resistance and recovery [22–25]. In addition to water deficit, rising temperatures and more frequent episodes of very high temperatures are expected due to climate change, which could increase ecosystems susceptibility to die-off events [26–29]. Anthropogenic pressure (e.g., overgrazing, agricultural intensification) may also increase the vulnerability of ecosystems [3,30,31], but a negative impact of human management on vegetation is not evident at the Sahelian scale [10,11,32,33]. Thus, given the crucial role of woody plant resources for the livelihoods of local populations [3], it is of great importance to understand how woody vegetation endures and recovers from drought events, and which factors are more important in modulating vegetation response.

The term "recovery" (from drought) can take on different meanings, such as the rate of return of an ecosystem towards its pre-disturbance state, or a post-disturbance increase in woody plants density, greenness, and/or canopy cover [16,24,34–37]. Here, we focused on the recovery rate of woody vegetation greenness and (water content in) aboveground biomass of the studied ecosystem, by analysing time series of optical and microwave remote sensing data. As increases in vegetation greenness/biomass can be related to either growth of new individuals or to the resprouting of individuals that survived the drought, we here considered both together.

At local, regional and global scales, remotely sensed data is a fundamental tool to understand dryland woody cover [38–40] and its trends [10,41,42]. However, indices frequently used to estimate vegetation dynamics, for example, the Normalized Difference Vegetation Index (NDVI) [43], typically fail to make a direct link between the remotely sensed observations and tree mortality/recovery. This happens mainly because NDVI estimates cannot differentiate between herbaceous and woody foliage production in drylands, as in those regions the grass layer is continuous and woody plants are sparse, resulting in NDVI values dominated by the grass layer dynamics [44,45]. New approaches based on phenology [16,46] rely on satellite sensors that do not cover the period of the Sahel droughts in the 1970s and 1980s (e.g., the Moderate Resolution Imaging Spectroradiometer, MODIS). However, recent drought events give us the opportunity to study the extent and drivers of woody vegetation mortality and recovery in more detail. Brandt et al. [16] used a phenology-based approach to decouple woody from herbaceous vegetation dynamics in the Senegalese Ferlo and identified an extensive increase in woody plants for the period 2000–2013, followed by a large scale negative woody cover anomaly in 2014–2015. Such large scale anomaly was found to be related to a widespread die-off mainly of *Guiera senegalensis*, a shrub species which is susceptible to drought, especially in semi-arid regions [16,47].

The die-off event in 2014–2015 was severe and it is now several years in the past, which allow us to use updated in situ information, new satellite data and sensors, and improved methodologies to revisit the event. As stressed by van der Molen et al. [20], vegetation response during a drought is largely understood, but a lack of knowledge remains for the months to years after its occurrence. Thus, the 2014–2015 die-off event offers a unique opportunity to carry out a comprehensive assessment of the woody vegetation response immediately after the die-off event (i.e., impact) and some years later (i.e., recovery). As a follow-up to Brandt et al. [16], we proposed to fill three knowledge gaps that still remain: (a) long-term trends in vegetation greening were already widely studied in West Africa [48], but inter-annual woody vegetation fluctuations need further understanding on the long-term. Here, novel long-term passive microwave data were used to study such fluctuations and compare them with previous woody vegetation anomalies, starting in the late 1980s. Besides covering a longer period than the dataset used by Brandt et al. [16], microwave data can provide more direct estimates of woody vegetation biomass, improving previous findings regarding the die-off event; (b) Although Brandt et al. [16] found signs of an early regeneration process during a field campaign in 2015, a longer observation period is necessary to properly quantify woody vegetation recovery after the disturbance. Thus, the longer time series of remotely sensed data now available allowed us to better estimate woody vegetation recovery; (c) Brandt et al. [16] discussed the biophysical mechanisms affecting woody vegetation dynamics, but a spatially explicit analysis to determine the relative importance of soil variables, human-pressure, and past biomass accumulation to woody plants mortality is still missing. Moreover, factors affecting post-disturbance recovery are yet to be evaluated.

In addition to the points previously raised, much is still unknown about the mechanisms leading to tree mortality at the ecosystem scale and about their resilience to drought events (e.g., Reference [20]). Therefore, we here aimed to advance towards a better understanding of woody vegetation response to drought by studying this specific drought-induced die-off case in Senegal. More specifically, we focused on answering the following questions:

1.  How severe was the 2014–2015 die-off event when compared to woody vegetation anomalies in the past three decades?
2.  Which areas in central and eastern Senegal were more severely impacted by the die-off, and which areas presented a higher relative recovery?
3.  How did edaphic characteristics, human pressure, and pre-drought vegetation dynamics affect the severity of the die-off? How did edaphic characteristics and human pressure affect the post-disturbance recovery?

To address these research questions, two types of remote sensing data were used: microwave data covering a long time range and optical data with a finer spatial resolution. We used recently-developed vegetation optical depth (VOD) datasets, which capture the plant water content and can be used to estimate inter-annual woody biomass variations [32,42,49]. The new long time series of Ku-band VOD data [49] allows characterizing recent die-off events in perspective with previous droughts and with the greening of the Sahel (question 1). Moving to a finer spatial resolution, optical satellite (MODIS) and field data (used to validate remotely sensed estimates), combined with a phenology-based approach, were used to identify spatial patterns of woody vegetation mortality and post-disturbance recovery (question 2). Finally, MODIS data together with ancillary data on soil and population density were used to determine which factors can influence the severity of woody vegetation mortality and recovery in the region (question 3).

## 2. Materials and Methods

### 2.1. Study Area

The study area is located in the western Sahel and covers central and eastern Senegal, which was affected by a drought-induced woody vegetation die-off in 2014–2015 (Figure 1). The climate in

the region is characterized by warm temperatures all year long (mean annual temperature around 29 °C) and no extreme temperature anomalies were observed in the last 20 years (see Figure S1 in Supplementary Materials). The region presents a short rainy season between June and October, with mean annual precipitation around 450 mm [16,50,51] (Figure 1b). Vegetation consists of a continuous herbaceous layer, which wilts towards the end of the rainy season, and sparsely distributed trees and shrubs [52]. The study area is part of the Ferlo region and mainly used for sylvo-pastoral activities [33,51]. Soils in the Ferlo region can be subdivided into sandy and ferrugineous (Figure 1d). Ferrugineous soils are characterized by an enrichment of low-activity clays in their subsoil and cover a large range of base saturation, corresponding with lixisols (when base saturation is high) or acrisols (when base saturation is low) [53]. Moreover, while the sandy soils are deeper, the ferrugineous soils are more shallow, due to the presence of an impervious laterite layer close to the surface, resulting in higher runoff in the latter [16,51].

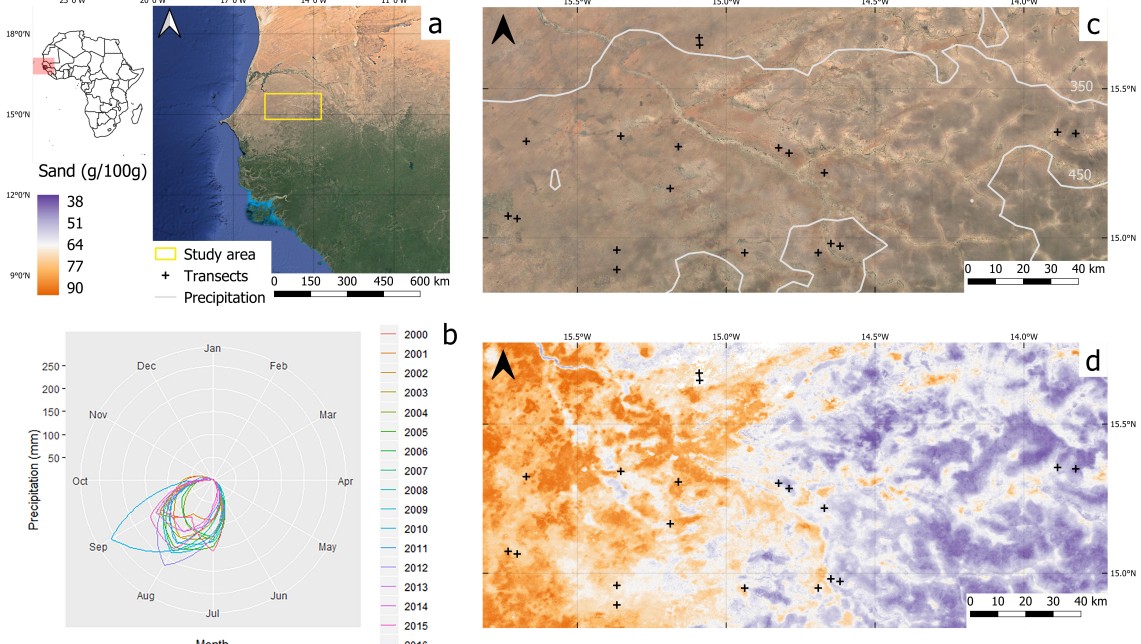

**Figure 1.** (**a**) Map of Senegal, with the study area highlighted in yellow [54]. (**b**) Polar seasonal plot of precipitation in the study area (2010–2016). The time series was restricted to allow visualization of possible seasonal variations in precipitation around the year of the die-off, which in fact are not observed. (**c**) Zoomed inset maps show a true-colour image [55] with precipitation isohyets (350–450 mm) and (**d**) sand content obtained from ISRIC World Soil Information [56]. While the western region has more sandy soils, the eastern region has shallow ferrugineous soils. The centre of the transects where field data were collected is shown in (**c**,**d**).

*2.2. Field Data*

The field campaign was conducted in September–October 2015, right after the die-off event. Field data sampling was performed in 19 transects of 500 m. A species survey and plant dimensions measurements (i.e., diameter, height and crown spread) were made for all woody individuals with more than 5 cm of basal stem diameter, and the number of dead individuals was noted, allowing for an estimation of the percentage of dead individuals per transect. The sampling was conducted inside three circular plots of around 20 m radius distributed along the 500 m transects, spaced at 250 m intervals (see Figure S2 in Supplementary Materials). Assuming that our sampling was able to capture a representative part of woody vegetation characteristics and variability inside a 500 × 500 m pixel, we related data collected on the field with indices derived from 500 m resolution MODIS data. All transects were revisited in August 2017.

### 2.3. Earth Observation Data

Three different types of satellite data were used to estimate vegetation dynamics: low and high-frequency VOD, and MODIS NDVI derived from optical reflectance data. While VOD data capture the water content of vegetation, which can be directly related to aboveground biomass [32,57–59], its spatial resolution is too coarse (i.e., 25 km) for spatially detailed analyses, making the inclusion of optical imagery necessary. Low-frequency SMOS-IC L-VOD data are able to directly assess the biomass of woody plants with little interference coming from green foliage [32,59,60]. We used the novel version 2 of the SMOS-IC product, which contains data for the period 2010–2018. The method used for the derivation of the SMOS-IC product focused on making soil moisture and VOD retrievals as independent as possible from ancillary data [60]. We aggregated all available SMOS-IC L-VOD images (for the period 2010-2018) into monthly median composites, and then used spatial-averages over the study area to create a time series of standing woody biomass estimates [10,59]. To test if the impact of the drought on standing biomass was detectable at a monthly time scale, we applied a breakpoint detection algorithm (BFAST) [61] on the L-VOD time series. BFAST decomposes time series into seasonal, trend and remainder components, while detecting abrupt changes. Here, the trend component is represented by a piecewise linear model.

Since L-VOD is only available for a short period, it cannot be used to relate the 2014 drought to previous events. Thus, we used the novel long-term Ku-band VOD (Ku-VOD) time series (1988–2017), with a spatial resolution of around 25 km, obtained by merging data from the SSM/I, TMI, AMSR-E, AMSR2, and WindSat sensors [49]. In contrast to the L-VOD, the high frequency of the Ku-band captures the water content of both the woody components and of the green foliage of the vegetation (i.e., it is also sensitive to the water content in the herbaceous layer). For that reason, we used Ku-VOD data from the dry season (i.e., the minimum value between October and May), avoiding interference from the grass layer and minimizing impact from woody foliage, thus mainly capturing the woody biomass. Although VOD retrievals are sensitive to both vegetation biomass and water stress [62], for the sparsely vegetated areas here studied we assumed a strong linear relationship between woody plant water content and biomass, supported by the findings of Fan et al. [2], Brandt et al. [10], Liu et al. [58], Rodríguez-Fernández et al. [59], and Tian et al. [63]. However, to further remove variations in the dry season VOD caused by inter-annual rainfall variability, following Brandt et al. [64], the following steps were taken:

- The peak growing season VOD (i.e., the maximum value between June and September) was extracted for each year and was used as a proxy of water balance conditions.
- The relationship between peak VOD and the minimum dry season VOD of the following dry season was assessed, and if a significant ($p < 0.05$) relationship was found, the coefficients of the regression were used to predict dry season VOD using the growing season peak.
- A reference VOD was predicted using the mean peak VOD for the entire time series.
- The predicted dry season VOD was then subtracted from the reference VOD, and this value was added to the minimum dry season VOD, resulting in a VOD value with almost no influence from rainfall and grass layer variability. Hereafter, we call this value simply the dry season VOD.

This process resulted in a time series with one year less because there is no growing season data available for 1987, which would be used to estimate the dry season Ku-VOD for 1988. We then summed dry season Ku-VOD values over the pixels covering the study area to obtain an approximation of the annual standing biomass. Differences between years are assumed to mostly reflect standing biomass gains and losses [32].

Finally, we used NDVI data from the MODIS MOD13A1 product, version 6 [65]. The spatial resolution of 500 m allows the identification of more detailed spatial patterns of vegetation dynamics for the period 2000–2018, and in addition, the field sampling was designed to be comparable with this resolution. We used 16-day NDVI estimates, with low-quality values filtered out by using the quality

data layer, and then aggregated to monthly maximum values. Moreover, the MCD64A1 Version 6 Burned Area product at 500 m resolution [66] was used to mask areas affected by fires.

## 2.4. Ancillary Environmental Datasets

Several environmental datasets were included to study their possible role in the drought-induced woody plant die-off and post-disturbance recovery, and thus helping to answer the third research question here proposed. Gridded data on soil variables (i.e., aluminium, nitrogen, phosphorus, cation exchange capacity, and sand) were acquired from the International Soil Reference and Information Centre (ISRIC) World Soil Information at a 250 m resolution [56]. Terrain slope data were derived from the NASA Shuttle Radar Topographic Mission elevation data [67], processed and provided by Fick and Hijmans [68] at 1 km. Human population density data in 2015 were obtained from the Gridded Population of the World version 4 Revision 10 dataset [69] at 1 km. SMOS L3 soil moisture data downscaled to a 0.05° (∼5.5 km in Senegal) spatial resolution were acquired for the period 2010–2015 [70]. By using dual polarization and multi-angular SMOS brightness temperature measurements, the L-band Microwave Emission of the Biosphere radiative transfer model is able to retrieve both VOD and soil moisture from SMOS data [60,71]. The overall per-pixel soil moisture condition was obtained by first calculating the annual growing season (June-October) average soil moisture, and then the average annual soil moisture for the stable period before the drought (i.e., 2010–2013). Reasons for selecting the period of 2010–2013 as the stable period are better explained in Section 2.7. The annual soil moisture anomaly was calculated by subtracting the annual growing season average from the long-term average of the whole period (2010–2015). All environmental datasets were resampled to a common resolution of 500 m using the nearest-neighbour method. Although being a simple method for image resampling, the nearest-neighbour method was sufficient in this case as nearby pixels are strongly correlated in the datasets used, and as the resolution of most of the used products is not too discrepant from the 500 m resolution used.

The CHIRPS dataset [72] provides monthly rainfall data at 5 km spatial resolution estimated from rain-gauge stations and satellite imagery, and was used here to represent rainfall conditions in our study area. More specifically, we used the standardized precipitation, calculated as the annual precipitation anomaly divided by the standard deviation. This allowed us to characterize the droughts in our study area since the 1980s, according to McKee et al. [73]. In this study, we only used rainfall and soil moisture data to better understand vegetation dynamics in our study area based on the assumption that vegetation growth in the Sahel is mainly limited by water availability [6,7,74], thus disregarding other climatic-variables.

## 2.5. Die-off Severity Index

To estimate the impact of drought on woody vegetation, we derived a Die-off Severity Index (DoSI) from MODIS NDVI data. To avoid interference of the grass layer and isolate the signal of woody plants, we used only NDVI values from December ($NDVI_{dec}$) for each year, as no annual herbaceous vegetation is green and most woody plants have full foliage in this month [38]. The DoSI shows the difference between the state of woody vegetation before the drought (i.e., mean $NDVI_{dec}$ for the stable period) and just after the drought event (Figure 2). To evaluate the performance of the DoSI, the percentages of dead individuals per transect measured in the field were correlated with the DoSI values overlaying the transects. Although drops in $NDVI_{dec}$ can happen due to, for example, loss of leaves and/or vegetation health, we here relate the specific case of $NDVI_{dec}$ drop in 2015 as a "woody vegetation die-off", due to its high correlation with woody vegetation mortality measured in the field (see Results). The DoSI was calculated by subtracting the NDVI representing the vegetation state right after the die-off ($NDVI_Y$) by the mean NDVI during the stable period before the drought ($NDVI_{stab}$) (Equation (1)).

$$DoSI = NDVI_{stab} - NDVI_Y. \tag{1}$$

Finally, to evaluate the impact of dead herbaceous mass on DoSI values, we used Google Earth's true colour images to visually select small regions (10 polygons with around 250 km$^2$ each) without woody vegetation. If the DoSI estimated for those regions is very low, we could assume that dry herbaceous mass does not have a substantial interference over the DoSI.

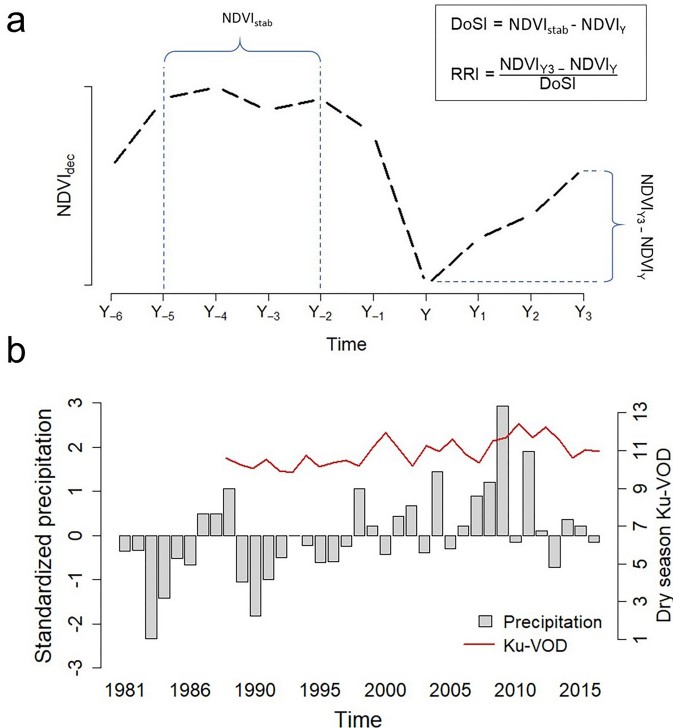

**Figure 2.** (**a**) Schematic time series highlighting periods and years used for the calculations of the DoSI and the RRI (the latter was adapted from Reference [75]). The post-drought woody vegetation state is represented by the time step "Y" (with Y corresponding to 2015), and the stable period before the drought (2010–2013) as "NDVI$_{stab}$". (**b**) Standardized precipitation, calculated as the annual precipitation anomaly divided by the standard deviation, and summed dry season Ku-VOD for the study area. The extreme and severe droughts in the 1980s and 1990s, respectively, the years of above-normal precipitation in 2008–2012, and the mild drought in 2014 are observable.

*2.6. Relative Recovery Indicator*

To access the recovery after the 2014–2015 die-off event, we used the Relative Recovery Indicator (RRI) as proposed by Frazier et al. [75]. The RRI accounts for both the amount of NDVI recovered after the disturbance and the degree of change in NDVI due to the disturbance (Figure 2a). It is calculated by subtracting the NDVI$_{dec}$ of the vegetation state right after the die-off ($NDVI_Y$) from the NDVI$_{dec}$ of the last year in the time series ($NDVI_{Y3}$) (i.e., the absolute recovery), and dividing the result by the value of the DoSI (Figure 2a; Equation (2)).

$$RRI = \frac{NDVI_{Y3} - NDVI_Y}{DoSI}. \tag{2}$$

It is noteworthy that using the absolute recovery to calculate the RRI provides a measure of how fast the ecosystem is recovering from the mortality event, rather than how close the ecosystem is to its state before the drought. In the next section, long-term precipitation dynamics in our study area are presented, and the years selected to represent the stable period, the post-drought vegetation state, and the absolute recovery are specified.

### 2.7. Long-Term Precipitation Dynamics

Since the 1980s, a number of periods with low rainfall were observed in the study area, but none was as pronounced as the one in 1983–1984 (Figure 2b), when an extreme drought, due to its severity and duration, struck the area [73]. In turn, the period from 2008 to 2012 (except 2011) was marked by above-average precipitation (Figure 2b). Right after this period, woody vegetation standing biomass was relatively stable (Figure 2b). By comparing the standard deviation of dry season Ku-VOD for periods starting in 2013 and going several years to the past, the period between 2010 and 2013 presented the lowest variability, and thus, this period was selected as the stable period (i.e., $Y_{-5}$ to $Y_{-2}$ in Figure 2a) before the drought of 2014. Thus, the vegetation state during the stable period was defined as the average $NDVI_{dec}$ for 2010–2013, thereby avoiding to include years when woody vegetation density in the study area was substantially increasing (Figure 2). The negative rainfall anomaly in 2014, a mild drought [73], was not as strong as the ones of the 1980s and 1990s (Figure 2b). The post-drought vegetation state (i.e., $NDVI_Y$ in 2a) was defined as the $NDVI_{dec}$ value for the year 2015, since many woody plants do not die immediately after a drought. Finally, the $NDVI_{dec}$ difference between 2018 and 2015 was used as the absolute recovery, in order to calculate the RRI (Figure 2a).

### 2.8. Environmental Determinants of Spatial Differences in Die-Off Severity and Recovery

To evaluate the relative contribution of human pressure, soil and terrain characteristics on the severity of the woody plant die-off and on the following recovery, we created linear models regressing DoSI and RRI with human population density, soil nitrogen (N), phosphorus (P), aluminium (Al), and sand content, soil cation exchange capacity (CEC), and terrain slope. The average growing season soil moisture (2010–2013) was also included in the models. Finally, the woody foliage accumulation before the die-off (estimated as the $NDVI_{dec}$ trend between 2000 and 2013) was included in the DoSI model. The flowchart in Figure 3 summarizes all datasets used.

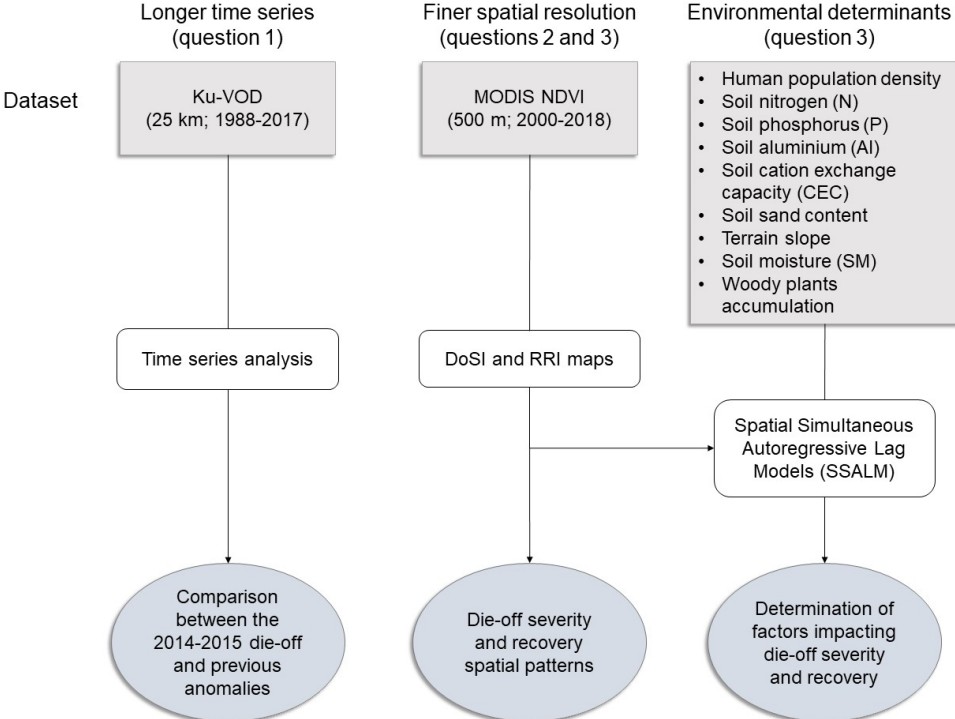

**Figure 3.** Flowchart summarizing datasets used and the outputs of each analysis. The research question(s) which relates to each dataset and/or analysis is also specified.

To test for spatial autocorrelation in the residuals of the fitted linear models, we ran a Moran's I test and evaluated the variogram of the residuals. The test indicated the presence of spatial autocorrelation in the residuals of both models ($p < 0.001$ for both), which can affect the estimates of the model coefficients. We then used a random sample of our dataset, with 12.7% (n = 12,000) of the total number of observations, to reduce the impact of the spatial autocorrelation in the linear models (and to reduce computation time as well). However, spatial autocorrelation in the residuals was still present, and the variogram pointed to a range of 7.95 and 7.75 km for the DoSI and RRI linear models, respectively. Thus, we used a Lagrange Multiplier Diagnostic for spatial dependence to find a suitable alternative for the linear model, which pointed to a Spatial Simultaneous Autoregressive Lag Model (SSALM) [76,77]. The SSALM contains a term for the spatial autocorrelation in the response variable ($\rho WY$), and the standard terms for $n$ explanatory variables ($X_n \beta_n$) and for the errors ($\epsilon$) (Equation (3)).

$$Y = \rho WY + X_1\beta_1 + X_2\beta_2 + ... + X_n\beta_n + \epsilon. \tag{3}$$

To create the neighbourhood structure, we defined all pixels closer than the range as neighbours, and assigned higher weights to closer neighbours in order to create a spatial weights matrix ($W$). Assigned weights ranged from 0.006 to 0.5. Adding the autocorrelation term indeed significantly improved the model (see Results).

Thus, we fitted two SSALMs to our data—one using the DoSI as the response variable (hereafter, the DoSI model) and the other model using the RRI (hereafter, the RRI model). For each explanatory variable in the DoSI model, a positive coefficient indicated that higher values of this variable are related to a more severe woody vegetation die-off, while negative values indicated a buffering effect. For the RRI model, positive coefficients indicated that such variables contributed to a faster relative recovery, while negative coefficients indicated the opposite.

## 3. Results

The results could be split into two main parts—first, we used VOD data to compare the 2014–2015 die-off with previous woody cover anomalies (question 1), and second, the finer spatial resolution of optical MODIS data was used together with ancillary datasets to study the spatial patterns and drivers of mortality and recovery (questions 2 and 3).

### 3.1. Long-Term Dynamics in Standing Biomass

The spatially-averaged L-VOD time series for our study area showed an abrupt drop in its trend component in 2014 (Figure 4a). After the drop, L-VOD started to increase again, indicating a recovery of woody vegetation with a subsequent increase in woody biomass. Soil moisture anomalies, strongly negative in 2014, pointed to a drought this year (Figure 4b). The longer Ku-VOD time series showed recurrent inter-annual fluctuations in annual minimum VOD in our study region (Figure 4c). The magnitude of such fluctuations differed greatly.

We used spatially summed dry season Ku-VOD as an indicator of the standing biomass of the area for a given year. Using the long-term average from the period 1989–2017 as a reference, the time series in Figure 5a showed strong losses in the 1990s, and a subsequent quasi-uninterrupted period of biomass accumulation starting in the 2000s. Besides the below-average years of 2003, 2007, and 2008, a stable period of high VOD values suggested that an accumulation of biomass occurred from 2009 to 2013 (when standing biomass reached levels 12.25% higher than the long-term average). After a year of well below-average rainfall in 2014 (Figure 5c), the total dry season Ku-VOD of the study area showed a considerable drop in 2015, decreasing to a level below the start of the monitoring period (1989). Considering the inter-annual deviations, the dry season Ku-VOD drop in 2015 was also the strongest loss over the period 1989–2017 (Figure 5b). This drop represents an 8% loss in total VOD from one year to the other. Another noticeable historical drop in VOD happened in 2002–2003, with a

7.67% loss in total VOD, while the anomalies in the early 1990s represent a 7.31% loss (comparing 1989 with 1994) (Figure 5).

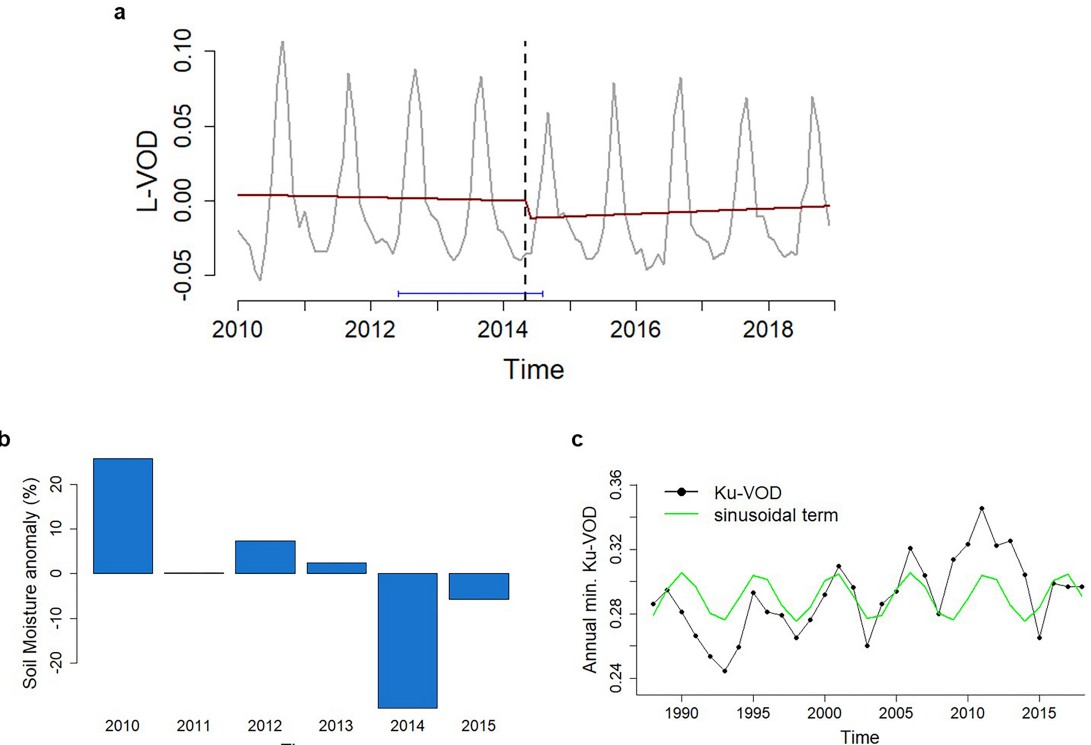

**Figure 4.** (**a**) L-VOD monthly time series as a proxy of the woody vegetation dynamics in the last 9 years at our study area. The fitted trend component (red) and the break in it (dashed line) are presented, with the respective confidence interval in blue. (**b**) Annual SMOS soil moisture anomaly, in percentages, spatially averaged for the study area. (**c**) Annual minimum Ku-band VOD spatially averaged for the study area, superimposed with a fitted sinusoidal term. Periodic drops in the annual minimum VOD are observed, highlighted by the sinusoidal term, suggesting an inter-annual cyclic pattern (see Discussion).

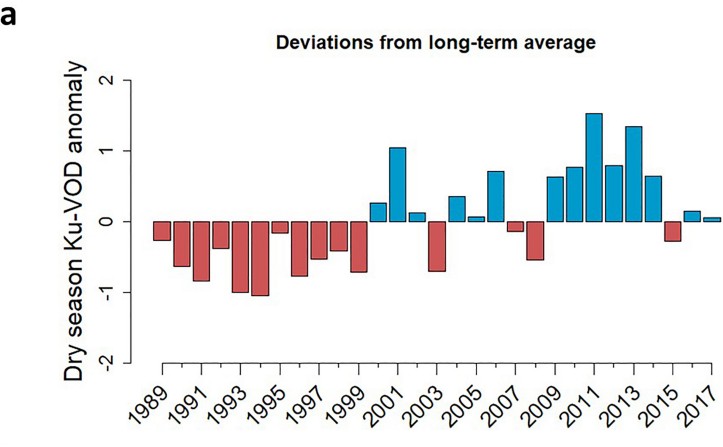

**Figure 5.** *Cont.*

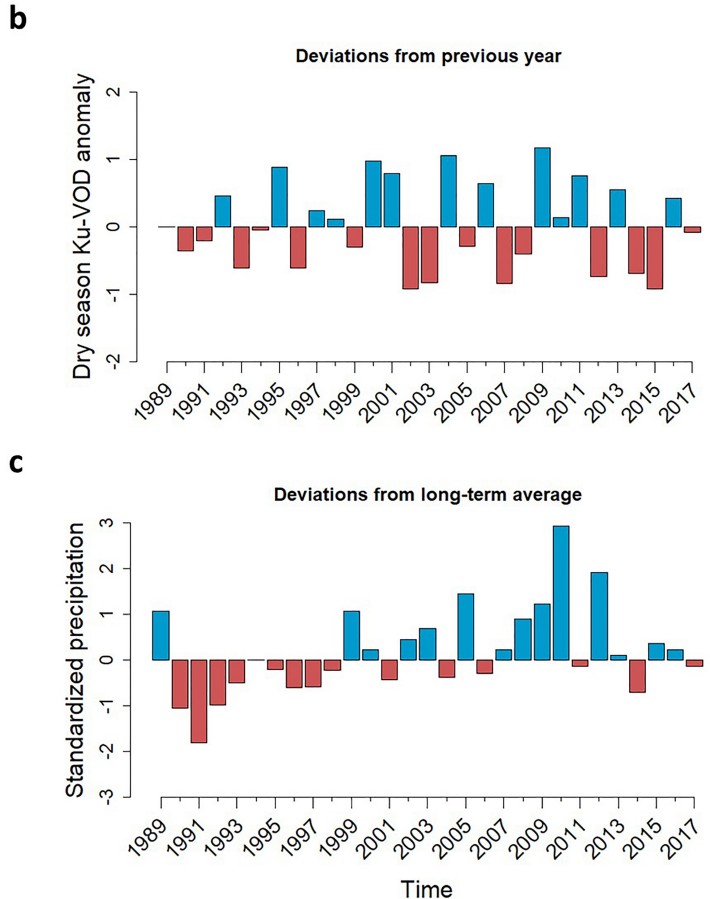

**Figure 5.** (**a**) Dry season Ku-VOD anomalies in relation to the long-term average (1989–2017) and (**b**) to the previous year. Total dry season Ku-VOD values were calculated by summing the values for all the pixels inside our study area and were used as an estimation of the standing woody biomass. (**c**) Standardized precipitation, calculated as the annual precipitation anomaly divided by the standard deviation.

### 3.2. Die-Off Severity and Recovery

The DoSI was strongly correlated with the percentage of dead woody individuals measured in the field (Pearson's correlation: $r = 0.832$, t = 6.01, $p < 0.001$; Figure 6). We used small regions without woody vegetation to assess the potential impact of dead herbaceous mass on the DoSI. Overall, the DoSI was very low in those sites (average of 0.02) when compared to the range of the DoSI over the region (0 to 0.28), indicating that dead herbaceous mass does not substantially interfere with the DoSI. The DoSI map shows higher die-off severity in the central-southern part and also in the northeast of the region (Figure 6a). A similar pattern is observed in the L-VOD data when the anomalies in 2015 in relation to the long-term average from the period 2010–2017 are calculated (see Figure S3 in Supplementary Materials). Although the coarse spatial resolution of the L-VOD data hampers a more detailed comparison, affecting its correlation with finer-scale products, we found relatively similar results when comparing the L-VOD anomalies in 2015 and the average DoSI values overlaying the L-VOD pixels (Pearson's correlation: $r = -0.555$, t = −3.66, $p < 0.001$). This means that more negative L-VOD anomalies were related to higher DoSI values, showing a consistency in the vegetation anomalies estimated from two different products. The RRI map shows a higher relative recovery towards the west and the south of the study area (Figure 6c). Moreover, areas that were affected by the die-off event but showed no recovery were mainly located in the northeast of the study area (Figure 6c).

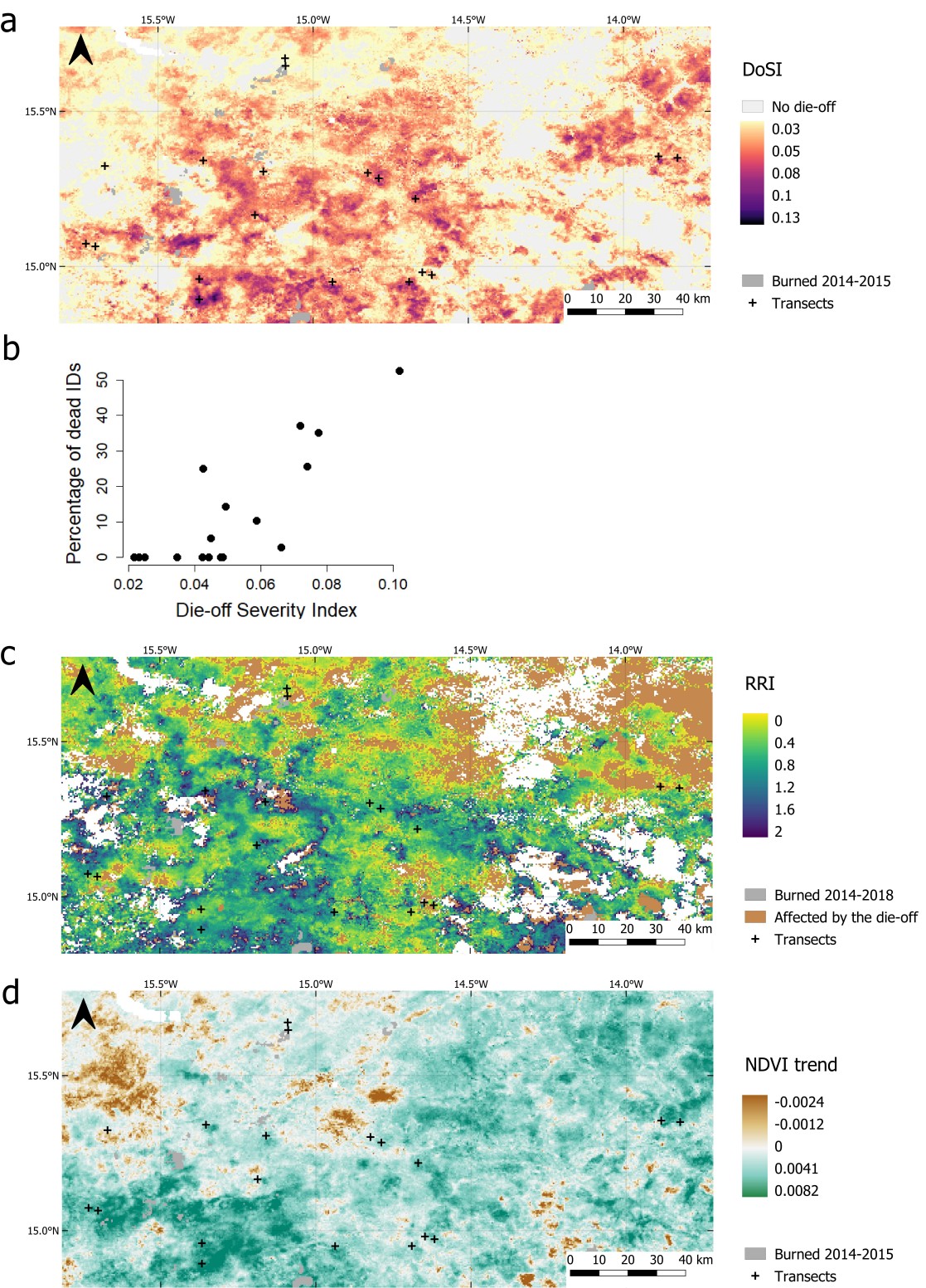

**Figure 6.** (**a**) The Die-off Severity Index (DoSI). Darker colours represent a higher die-off severity. Negative DoSI values were grouped as "No die-off". (**b**) Relationship between the percentage of dead woody individuals (IDs) measured in the field and the DoSI. (**c**) The Relative Recovery Indicator (RRI). Pixels within the yellow-blue colour ramp were affected by the die-off event but presented signs of recovery. Darker colours represent areas where woody vegetation recovery, relative to the level of mortality, was higher. Brown pixels were affected by the drought but did not show signs of recovery. (**d**) $NDVI_{dec}$ trend between 2010 and 2013, representing areas with high/low woody vegetation accumulation before the die-off.

### 3.3. Factors Impacting Die-Off Severity and Recovery

The SSALM results indicated that adding the autoregressive parameter improved both the DoSI model ($\rho$ = 0.964; LR test value = 5844.9; $p < 0.001$) and the RRI model ($\rho$ = 0.921; LR test value = 3156.0; $p < 0.001$), when compared to linear models. Sand content and soil moisture were the variables that had the most negative impacts on variations in the DoSI (Table 1). Soil P also showed a negative, although weaker, influence over the DoSI. The NDVI$_{dec}$ trend before the die-off (Figure 6d) was the variable which most positively impacted DoSI variations (Table 1). Soil aluminium concentration (Al) and terrain slope also affected positively the DoSI (Table 1). Human population density and soil N had no significant impact on the DoSI. Sand content was of high importance positively affecting RRI, followed by soil moisture and soil nutrients (N and CEC). The terrain slope showed a negative relationship with the RRI, and human population density had no significant impact (Table 1).

**Table 1.** Results from the Simultaneous Autoregressive Lag Models (SSALMs). Input values were scaled to make the coefficients comparable. Positive values indicate that an increase in the variable resulted in a higher DoSI or RRI, while negative values indicate the opposite. SM denotes soil moisture. Only coefficients for variables with $p < 0.05$ are shown.

| Coefficient | DoSI Model | | | | RRI Model | | | |
| --- | --- | --- | --- | --- | --- | --- | --- | --- |
| | Estimate | Std. Error | z-Value | *p*-Value | Estimate | Std. Error | z-Value | *p*-Value |
| Population | − | − | − | − | − | − | − | − |
| SM | −0.160 | 0.011 | −14.6 | <0.001 | 0.035 | 0.014 | 2.49 | 0.013 |
| Al | 0.054 | 0.007 | 7.3 | <0.001 | − | − | − | − |
| N | − | − | − | − | 0.055 | 0.01 | 5.38 | <0.001 |
| P | −0.031 | 0.008 | −4.12 | <0.001 | − | − | − | − |
| CEC | − | − | − | − | 0.06 | 0.018 | 3.42 | <0.001 |
| Sand | −0.102 | 0.015 | −6.97 | <0.001 | 0.1 | 0.019 | 5.24 | <0.001 |
| Slope | 0.065 | 0.007 | 8.71 | <0.001 | −0.058 | 0.009 | −6.11 | <0.001 |
| NDVI trend | 0.197 | 0.007 | 27.13 | <0.001 | NA | NA | NA | NA |

## 4. Discussion

Our study indicates that a rapid accumulation of woody vegetation biomass is not always a sign of a stable woody plants recovery, adding further depths to our understanding as compared to simplified views of the "greening Sahel". Prolonged above-average rainfall levels may allow woody biomass accumulation, mainly of drought-intolerant species which are immediately affected by negative rainfall anomalies, impacting vegetation production and standing biomass. Indeed, the 1983–1984 drought in the Sahel was exceptional [78,79]: it lasted two years and resulted in considerable precipitation and soil water deficits, which caused a mass dying of woody plants. Since then, years with negative rainfall anomalies occurred on a recurrent basis (Figures 2b and 5c), but with smaller magnitudes and shorter duration, and no cases of large scale mass dying were reported until the 2014–2015 event [16]. Moreover, recent research in the Sahel using vegetation optical depth (VOD) found recurrent highs and lows in the time series of satellite-estimated vegetation data starting in the 1990s [10]. This may suggest that the woody vegetation dynamics observed in this study are not showing a recovery from the Sahelian drought in the 1980s, as stated by Brandt et al. [16], but are part of a recurrent pattern of positive and negative woody vegetation anomalies with varying severity (Figure 4c ). Although the causes of such patterns still have to be evaluated, Brandt et al. [10] suggested that they may relate to rainfall anomalies driven by sea surface temperature variations [17]. Moreover, the temporal scale and spatial extent of such woody vegetation recurrent anomalies are also yet to be evaluated.

The abrupt drop in L-VOD around 2014 (Figure 4a) indicates a clear and considerable loss of aboveground biomass [32,59]. Field work has shown that this can be attributed to the loss of woody plants. The 2014–2015 anomaly was not extraordinary concerning annual rainfall and soil moisture, but due to the fact that several years with high rainfall had caused an accumulation of woody biomass

that was exceptional for this area and for the last 30 years. Most of the observed woody biomass accumulation was due to the establishment of a dense woody vegetation cover, mainly consisting of *G. senegalensis* and *Balanites aegyptiaca* [16], which were also the species most frequently sampled in our field campaign (see Figure S4 in Supplementary Materials). The former species was severely impacted by the 2014 drought, triggering a mass die-off on the shallow ferrugineous Ferlo soils and causing a drop in biomass levels of the study area below the level of 1989 (Figure 5a). The biomass loss was unique for this area over the studied period (1989–2017)—total VOD in the study area dropped 8% after the mild drought in 2014, while the losses after the more severe droughts in the early 1990s were of 7.31%. Unfortunately, it is not possible to make a comparison with the 1983–1984 drought due to a lack of longer-term data.

It is noteworthy that a quantification of the aboveground biomass losses using VOD data is not straightforward, as VOD is sensitive to both vegetation biomass and water stress [62]. However, we removed part of the inter-annual rainfall variability influence over dry season VOD values by following the method proposed by Brandt et al. [64]. In addition, evidence from the field campaign allows us to infer that a major part of the dry season Ku-VOD abrupt decrease in 2015 (Figure 5a,b) was due to the woody plants die-off, which directly relates to biomass losses. Finally, precipitation in 2015 was slightly above the average after the 2014 drought, which did not lead to an increase in dry season Ku-VOD as would be expected if variations in Ku-VOD for the studied case would be driven by relative water content. Contrary, dry season Ku-VOD in that year was very low (Figure 5). Another potential constraint refers to the use of field data for the validation of the die-off severity estimates from remote sensing data, as there is a spatial gap between both. To solve this issue we used a sampling design that tries to capture a great part of the variability inside a 500 m MODIS pixel. Although we opted to use 500 m spatial resolution data to assess the die-off severity over a larger extent, we suggest that future research could use finer resolution data to better illustrate relationships between the die-off and environmental/anthropogenic factors.

The widespread woody vegetation mortality was also evidenced by the Die-off Severity Index (DoSI), an indicator of negative anomalies in woody vegetation proposed in this study. The severity of the die-off was not spatially homogeneous, being higher in the central-southern and north-eastern parts of the study area (Figure 6a). This pattern could not be explained by anthropogenic pressure, but mainly by the pre-drought woody biomass accumulation. The $NDVI_{dec}$ trend between 2010–2013, used here as an indicator of the woody vegetation accumulation before the drought, was the variable that exerted the highest positive influence over variations in the DoSI (Table 1). This also supports the fact that the above-average rainfall conditions prior to the drought, which resulted in the woody biomass accumulation, made the ecosystem susceptible to a widespread woody plants die-off. Two possible causes for this pattern, which likely acted in parallel, are: (a) the biomass accumulation was mainly due to the growth of woody species that are not tolerant to droughts and were severely struck by the 2014 rainfall anomaly [16,34]; and (b) an intensification of soil water overdraft occurred due to higher evapotranspiration rates by the accumulated woody biomass, depleting water stored in the soil and leading to higher mortality levels [80,81]. The regular decrease in the soil moisture conditions observed by SMOS over the area during 2010–2014 (Figure 4b) is consistent with the latter hypothesis. Also in agreement with the latter hypothesis, Jump et al. [82] summarized case studies with similar patterns of biomass accumulation followed by a drought-induced die-off in other biomes.

Soil texture also presented a strong influence over the die-off severity (Table 1). The deep sandy soils in part of the study area are capable of retaining rainfall water for longer periods and allow the vegetation to develop deep roots, while the shallow ferruginous soils (identified by high Al and low sand content in the study area) present an impervious laterite layer and high runoff [16,51]. The higher severity of the die-off in the shallow ferruginous soils was evident in the model results (Table 1) and also visually clear from field observations (Figure 7). These findings are consistent with Tappan et al. [51], who found indications of large scale tree and shrub mortality on ferruginous soils following the 1983-1984 drought. Terrain slope also influenced die-off severity (Table 1), and some species were more

affected by the die-off than others. For example, the majority of dead individuals were *G. senegalensis*, while *B. aegyptiaca* withstood the dry year and *Combretum glutinosum* suffered with the drought but rapidly recovered [16]. This is consistent with the findings of Poupon and Bille [83], who suggested that topography has an important role on woody plants survival, and reported high mortality rates for *G. senegalensis* (40–63%) after a drought event in the Senegalese Ferlo in the 1970s, while *B. aegyptiaca* presented lower rates (5.5%). It is noteworthy that traits related to species-specific drought tolerance (e.g., hydraulic traits), not directly considered in our analysis, are important determinants of woody plants survival during drought [34]. Thus, better than considering species diversity alone, accounting for the diversity of plant functional types is fundamental for the assessment of ecosystems resilience. Finally, another mechanism that may have contributed to the woody plants widespread mortality is the root-shoot allocation trade-off [84,85]. When conditions were favourable, if plants allocated more resources to shoots and less to root growth, this could have led to an increased vulnerability to periods with low water availability [19,84]. In addition, increases in shoot growth can result in increased evapotranspiration [84], depleting water stored in the soil, as aforementioned.

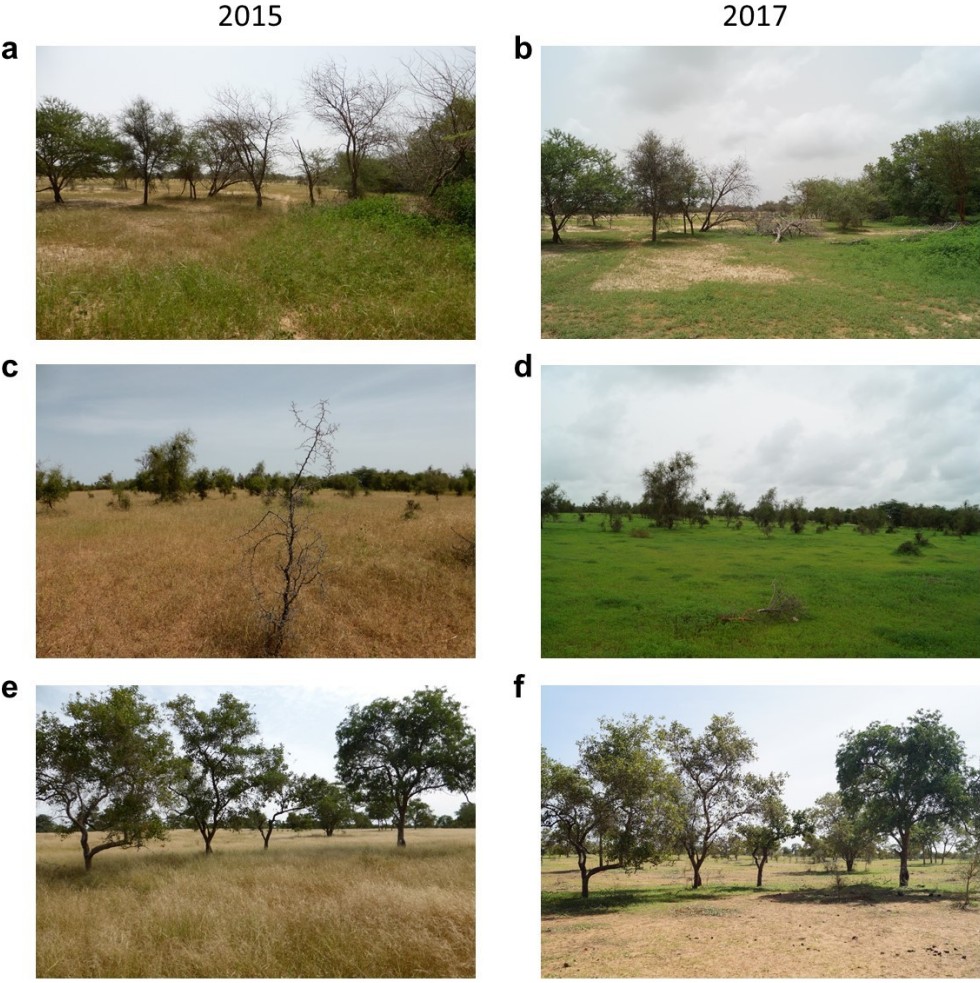

**Figure 7.** Photos of three transects sampled during the field campaign of 2015 and revisited in 2017. (**a**,**b**) Transect 9 was located in shallow ferruginous soils. Two big *G. senegalensis* individuals that appeared dead in the 2015 photo were not present anymore in 2017 (it is a common practice in the region to collect wood from dead trees and shrubs), while the acacia and *B. aegyptiaca* individuals remained. (**c**,**d**) Transect 16 and (**e**,**f**) transect 13 were located in more sandy soils and the lower amount of dead woody individuals in those transects is clear. Although a visual assessment of the recovery after two years should be treated with caution, the growth of new *G. senegalensis* and *B. aegyptiaca* individuals is already observed in (**f**).

Using remotely sensed data up to 2018 (three years after the die-off), we derived the Relative Recovery Indicator (RRI) to identify regions presenting faster post-disturbance recovery. The highest RRI values were observed in parts of the study area where soils were more sandy and where they were richer in nutrients, indicating that the low nutrient content and low capacity to retain water in shallow ferrugineous soils hamper a rapid recovery. It is well-known that water is the main constraint of vegetation growth in drylands, but soil nutrients (besides of co-varying and presenting co-limitation with water) can contribute to vegetation establishment and growth when water is sufficiently available (e.g., when rainfall conditions return to normal after a drought) [86,87]. Understanding how water availability and soil nutrients buffer woody vegetation against droughts is pivotal for ecosystems characterized by recurrent negative rainfall anomalies and, moreover, this suggests that attention has to be given to other variables than water availability to fully assess the resilience of dryland ecosystems. Such understanding gains even more importance considering that increased temperatures and decreased soil moisture—a possible scenario due to climate change [88,89]—may decrease soil nutrients availability [90].

## 5. Conclusions

In this study, we used novel long-term passive microwave data to compare events of negative woody vegetation anomalies in Senegal from 1989 to 2017, and optical data together with a simultaneous autoregressive model to determine the drivers of distinct die-off severity and recovery spatially. We found evidence of soil physical parameters impacting on how woody vegetation withstands and recovers from drought-induced die-off events in the Senegalese drylands. Given that negative rainfall anomalies are a recurrent phenomenon in the Sahel, monitoring woody vegetation dynamics in relation to climate change is of substantial importance, both for the conservation of local ecosystems and for the well-being of human populations in the region [3]. Although the negative rainfall anomaly in 2014 was much weaker than the ones in the 1980s and early 1990s, it caused a die-off which represented the highest inter-annual drop in dry season VOD for the studied period. The accumulation of woody plants during a pre-drought wet period was offset by the die-off, locally decreasing the standing woody biomass back to a level below the one of 1989. This illustrates how a greening trend does not necessarily mean that the ecosystem is recovering in a stable manner. A rapid accumulation of woody biomass can make an ecosystem vulnerable to a future die-off event during years of below-average rainfall, pointing to the need for a more complex understanding of the "greening Sahel". Therefore, we here suggest that, besides water availability, attention has to be given to other variables (e.g., soil nutrients and rapid biomass accumulation) to fully understand resilience in dryland ecosystems. As the interactions between past rainfall conditions, vegetation dynamics, and edaphic conditions can exert influence over the stability of ecosystems and livelihoods in drylands, better understanding them should support the achievement of some of the Sustainable Development Goals proposed by the United Nations, such as (1) No poverty, (2) Zero hunger, (3) Good health and well-being, and (15) Life on land [91].

**Supplementary Materials:** The following are available online at http://www.mdpi.com/2072-4292/12/14/2332/s1, Figure S1: Land surface temperature annual anomalies in the study area, for the period 2000–2018. Figure S2: Illustration of the field sampling design. Figure S3: L-VOD 2015 anomaly in relation to the long-term average. Figure S4: Proportion of species sampled and number of dead/alive woody individuals per transect.

**Author Contributions:** Conceptualization and methodology, P.N.B., M.B., W.D.K., R.F., B.S., and J.V.; software, P.N.B.; validation, P.N.B.; formal analysis, P.N.B., M.B., W.D.K., S.H., and I.S.; investigation, P.N.B., M.B., W.D.K., S.H., I.S., and J.-P.W.; data curation, P.N.B. and I.S.; writing—original draft preparation, PNB; writing—review and editing, all the authors; visualization, P.N.B.; supervision, B.S. and J.V.; project administration, W.D.K., S.H., R.F., B.S., and J.V.; funding acquisition, B.S. All authors have read and agreed to the published version of the manuscript.

**Funding:** This research was funded by the Belgian Science Policy Office in the framework of the STEREOIII program (project U-TURN, Understanding Turning Points in Dryland Ecosystem Functioning, grant SR/00/339). RF acknowledges the funding from the Danish Council for Independent Research (DFF), grant 6111-00258.

**Acknowledgments:** The authors would like to thank M. Mbaye and everyone else who supported the field data collection.

**Conflicts of Interest:** The authors declare no conflict of interest.

**Abbreviations**

The following abbreviations are used in this manuscript:

| | |
|---|---|
| NDVI | Normalized Difference Vegetation Index |
| MODIS | Moderate Resolution Imaging Spectroradiometer |
| VOD | vegetation optical depth |
| ISRIC | International Soil Reference and Information Centre |
| SMOS | Soil Moisture and Ocean Salinity |
| BFAST | Breaks For Additive Season and Trend |
| SSM/I | Special Sensor Microwave/Imager |
| TMI | Tropical Rainfall Measuring Mission |
| AMSR-E | Advanced Microwave Scanning Radiometer - Earth Observing System |
| AMSR2 | Advanced Microwave Scanning Radiometer 2 |
| CHIRPS | Climate Hazards Group InfraRed Precipitation with Station data |
| DoSI | Die-off Severity Index |
| RRI | Relative Recovery Indicator |
| SSALM | Spatial Simultaneous Autoregressive Lag Model |

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
