# Peer review of "Uncovering Dryland Woody Dynamics Using Optical, Microwave, and Field Data—Prolonged Above-Average Rainfall Paradoxically Contributes to Woody Plant Die-Off in the Western Sahel"

_remotesensing, doi:10.3390/rs12142332_

Round 1
Reviewer 1 Report
Oponent’s opinions to the article of Bernardino, P.N. et al. Uncovering dryland woody dynamics…
The article is built up well. The actual scientifical prpblem is well overthinked and good written down. I do not see any serious problem that could interfere to present it in journal Remote Sensing. The authors well the data in different spatial resolution and use properly the passive microwave data, that results about I haven’t heard too much before. I can find answers to every questions that I thought.
They are thinking of the complexity of the drought through the different databases and the necessity of the validation. The analyse the effect of one year (2014) in the time series after seeing also the other years. This extremity will be characterised worldwide in the near future, that is why the method can be used not only here.
Some minor oppinions:
- I would see better where is the study area on the Fig.1. The section ’a’ can be more overviewed.
- I think the equations should be in the text in ch. 2.5. and not only on the figure 2.
- For the readers would be better if the environmental determinant were in the list format in the ch. 2.8. and not only in the text.
- Why has not got title the fig. 3/c, when 3/a and 3/b have it?
- You write in the text (ch. 3.2., line 318) that the data range of DoSI in the study area is from 0 to 0.28, but on the legend of Fig. 5/a I see only from 0 to 0.13.
- The Fig.5. is too wide because of the ’b’ section. You should re-plan it.
- The photos on Fig.6. can be larger a little.
- There are some extra space in the text of the References; fi. line 498, 505.
Reviewer 2 Report
see attached file
